# Direct and indirect effects of land-use intensity on plant communities across elevation in semi-natural grasslands

Oksana Y. Buzhdygan[1]*, Britta Tietjen[1,2], Svitlana S. Rudenko[3], Volodymyr A. Nikorych[4], Jana S. Petermann[2,5]*

1 Freie Universität Berlin, Institute of Biology, Theoretical Ecology, Germany, Berlin, Germany, 2 Berlin-Brandenburg Institute of Advanced Biodiversity Research (BBIB), Berlin, Germany, 3 Department of Ecology and Biomonitoring, Chernivtsi National University, Chernivtsi, Ukraine, 4 Department of Agrotechnologies and Soil Science, Chernivtsi National University, Chernivtsi, Ukraine, 5 Department of Biosciences, University of Salzburg, Salzburg, Austria

* oksana.buzh@fu-berlin.de (OYB); jana.petermann@sbg.ac.at (JSP)

**Data Availability Statement:** The data used to support the conclusions of this study are submitted to Dryad (DOI: 10.5061/dryad.n8pk0p2rx).

## Abstract

Grassland biodiversity is vulnerable to land use change. How to best manage semi-natural grasslands for maintaining biodiversity is still unclear in many cases because land-use processes may depend on environmental conditions and the indirect effects of land-use on biodiversity mediated by altered abiotic and biotic factors are rarely considered. Here we evaluate the relative importance of the direct and indirect effects of grazing intensity on plant communities along an elevational gradient on a large topographic scale in the Eastern Carpathians in Ukraine. We sampled for two years 31 semi-natural grasslands exposed to cattle grazing. Within each grassland site we measured plant community properties such as the number of species, functional groups, and the proportion of species undesirable for grazing. In addition, we recorded cattle density (as a proxy for grazing intensity), soil properties (bare soil exposure, soil organic carbon, and soil pH) and densities of soil decomposers (earthworms and soil microorganisms). We used structural equation modelling to explore the direct and indirect effects of grazing intensity on plant communities along the elevation gradient. We found that cattle density decreased plant species and functional diversity but increased the proportion of undesirable species. Some of these effects were directly linked to grazing intensity (i.e., species richness), while others (i.e., functional diversity and proportion of undesirable species) were mediated via bare soil exposure. Although grazing intensity decreased with elevation, the effects of grazing on the plant community did not change along the elevation gradient. Generally, elevation had a strong positive direct effect on plant species richness as well as a negative indirect effect, mediated via altered soil acidity and decreased decomposer density. Our results indicate that plant diversity and composition are controlled by the complex interplay among grazing intensity and changing environmental conditions along an elevation gradient. Furthermore, we found lower soil pH, organic carbon and decomposer density with elevation, indicating that the effects of grazing on soil and related ecosystem functions and services in semi-natural grasslands may be more

**Funding:** BOY: POINT fellowship. Dahlem
Research School, Freie Universität Berlin. RSS:
0103U001966, 2004-2007 and 0107U001245,
2007-2008. The State Fund for Fundamental
Research of Ukraine. PJS: Open Access Publication
Fund of the University of Salzburg. The funders had
no role in study design, data collection and
analysis, decision to publish, or preparation of the
manuscript.

**Competing interests:** The authors have declared
that no competing interests exist.

pronounced with elevation. This demonstrates that we need to account for environmental
gradients when attempting to generalize effects of land-use intensity on biodiversity.

## Introduction

Grasslands cover more than 40% of the global terrestrial area [1] and are among the most spe-
cies-rich habitats in Europe [2]. As a result of their high biodiversity, grasslands provide high
yield and quality of forage [3] and deliver crucial ecosystem functions and services beyond that
of livestock forage production [4–6], for example pollination [7], carbon storage [8–10], wild-
life habitat provisioning [11–18], soil erosion control, and water flow regulation [19, 20]. How-
ever, grassland biodiversity compared to the diversity in other ecosystem types is among the
most vulnerable to human impact, particularly to land-use change [21]. Of extraordinary
importance for biodiversity are semi-natural grasslands [2, 13, 17, 18], which are remnants of
habitats created by tree cutting, haymaking, or low-intensity traditional farming [20]. In order
to survive and to function, semi-natural grassland communities require regular grass removal,
for example through grazing [2]. However, the effects of grazing on grassland biodiversity
have been found to depend on the land-use intensity [2, 22] and on particular environmental
conditions [23], yielding contrasting results. The inconsistency in the observed patterns may
depend upon the balance of different mechanisms underlying the relationships among grazing
and plant community composition and diversity along environmental gradients (the hypothe-
sized mechanisms derived from literature are summarized in S1 Table in S1 Appendix and
Fig 1).

The direct impact of cattle on a plant community (Fig 1, *path 1*) falls into several different
categories [24], and includes mechanical, chemical [25], and biological [13] effects on vegeta-
tion (S1 Table in S1 Appendix). Grazing can impede as well as improve local colonization pro-
cesses by plants for example through reduction of propagules of extant species or by increasing
the dispersal of propagules of new species to a site [26]. Additionally, grazing can increase
plant diversity by reducing competition via direct consumption of competitively dominant
plant species [23]. By contrast, if grazing promotes species dominance by increasing the abun-
dance of grazing–resistant, unpalatable species, then resource availability for other plant spe-
cies decreases, thus reducing biodiversity [23]. In parallel, grazing can increase the spatial
heterogeneity of vegetation [27], in comparison to e.g. hay meadows, which in turn could ben-
efit plant diversity.

The indirect effects of cattle grazing on the plant community can be mediated by changes
in soil-related parameters (Fig 1, *paths 3* and *8*), for example, soil compaction [25, 28], which
results in bare soil exposure and restricts the ability of roots to penetrate and of shoots to
emerge [25], therefore preventing the restoration of vegetation [29]. Furthermore, soil com-
paction may affect the availability of water and nutrients to plants [28, 30] (Fig 1, *path 5* and
*path 8*) through the altered decomposition processes via reduced soil porosity [28] (Fig 1, *path
10*) or decreased activity of soil decomposers [31] (Fig 1, *path 6*), which in turn are known to
impact nutrient cycling and, thereby, resource availability for plants [32] (Fig 1, *paths 5*, *7* and
*8*). On the other hand, the deposition of excrements by cattle in soil can relax soil acidification
[25] (Fig 1, *path 9*), and, as a result, can increase the nutrient uptake by plants [28, 33] (Fig 1,
*paths 8*). Furthermore, cattle manure serves as the resource input for soil decomposers [30]
(Fig 1, *path 4*), therefore resulting in increased soil organic matter content [34] (Fig 1, *path 7*),
and ultimately in greater resource availability to plants (Fig 1, *paths 8* and *5*). Moderate levels

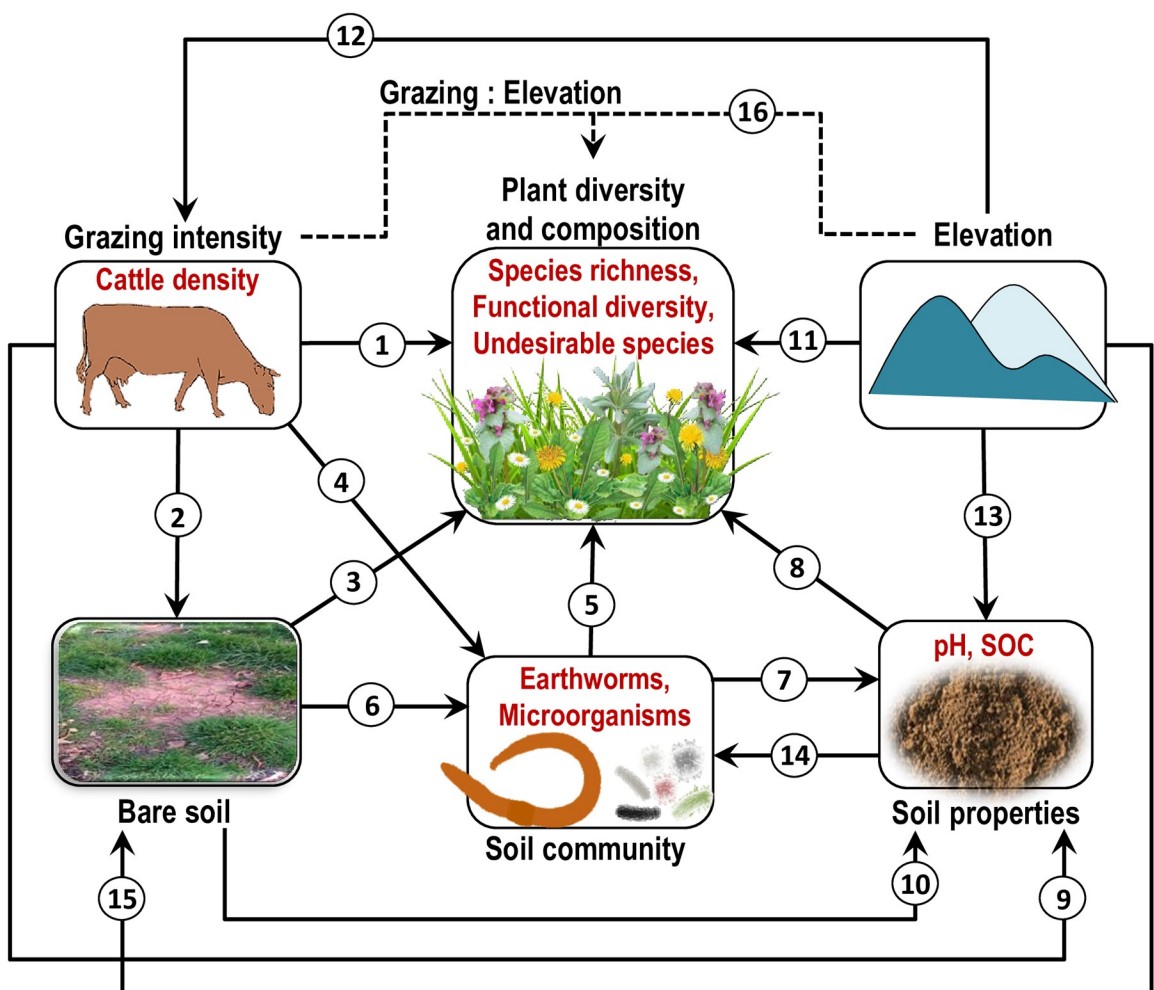

**Fig 1. Conceptual model of the expected causal direct and indirect effects of grazing intensity and environmental conditions on plant diversity and composition.** The conceptual model is based on hypotheses derived from the literature (see S1 Table in S1 Appendix for hypotheses and references). SOC, soil organic carbon. Dashed path 16 shows an interactive effect of grazing and elevation.

of soil disturbance by trampling can also increase water and nutrient availability in vegetation gaps [26] and may stimulate germination of plants from the soil seed bank as a result of increased light availability and nutrient-rich and pathogen-free soil [27] (Fig 1, *path 3*).

In addition to the direct and indirect effects that grazing may pose on the plant community, grazing effects may depend strongly on the environmental context, for example on elevation, topography, precipitation, soil disturbance, and site productivity [35, 36]. Grazing patterns may vary with elevation (Fig 1, *path 16*) because higher elevations are used later in the season than low elevations [37]. Also, livestock density is usually lower at higher elevation (Fig 1, *path 12*) leading to lower net effects of grazing on the grassland community. On other hand, an extensive erosion processes in steep slopes of terrain with heavy runoff and soil losses limit water and nutrient availability to plant growth [38]. Thus, the use of vegetation by livestock is expected to have more pronounced effects on plant diversity and composition in uplands in contrast to those in lowlands [39] (Fig 1, *path 15*). Furthermore, soil types vary with topography (S1 Fig in S1 Appendix). This might alter soil properties (i.e., pH and soil organic carbon) along the elevation gradient [40] (Fig 1, *path 13*), therefore influencing nutrient availability to

plants (Fig 1, *paths 8, 15* and *5*). In addition, higher soil leaching with increasing elevation is also expected to increase soil acidity and decrease soil organic carbon [40]. On the other hand, increased elevation involves altered climate conditions (i.e., increased solar radiation, precipitation, humidity, extensive wind exposure, and reduced air temperature), which may further affect local vegetation by shaping the regional species pool composition, i.e. via filtering of species physiologically capable of living under these environmental conditions [41] (Fig 1, *path 11*). High topographic variability linked to steep altitudinal gradients across mountain areas are found to result in greater habitat diversity [42], and therefore may lead to increased plant species richness along the elevation gradient (Fig 1, *path 11*). Increased precipitation with increasing elevation in the Carpathians [43] may also alter the effects of grazing on plant diversity [44].

The variety and interacting effects of these different mechanisms may be responsible for the idiosyncratic nature of grazing effects on plant communities in grasslands. Given that a single mechanism may produce multiple patterns (S1 Table in S1 Appendix), while multiple mechanisms may lead to convergent patterns or trade-offs among different effects, it has been shown that the relationships among grazing and biodiversity are best understood within the context of multivariate models [45]. In fact, recent studies have started to reveal the variety of pathways via which grazing affects plant community in grasslands [23, 46–48]. However, the majority of the existing evidence is limited to contrasting grazed *vs* ungrazed (i.e., fenced) systems. Studies on the effects of realistic grazing situations, where grazing managements varies from low to high intensities, are largely under-represented. This limits our ability to predict grazing effects on natural or semi-natural grasslands where fencing is uncommon.

In this study, we evaluate the simultaneous direct and indirect effects of grazing intensity and elevation on plant species richness, number of key plant functional groups (legumes, grasses (Poaceae), other monocots (rushes and sedges), and forbs), and on the proportion of species undesirable for grazing in semi-natural grasslands at large topographic scale ranging from the Carpathian Mountains in Ukraine, across the adjacent foothills to the plain areas (Fig 2), which is a largely under-represented region in scientific literature. We test concomitantly the following questions: (*1*) Do the effects of grazing intensity and elevation on plant community operate via changes in soil properties and altered soil biota? (*2*) Does grazing impact on plant diversity varies across the elevation gradient (i.e., tested with an interactive effect of grazing and elevation, Fig 1, *path 16*)? (*3*) Is plant community composition affected by elevation, cattle density, soil properties, and soil biota? To address questions 1 and 2 we use structural equation modelling (SEM), which allows testing simultaneous influences of multiple factors from observational data in complex systems and to distinguish direct and indirect effects of these factors [49]. Fig 1 shows the conceptual graph underlying our study and, thus, the SEM model. To test question 3 we use nonmetric multidimensional scaling (NDMS).

## Material and methods

### Study area

We studied 31 semi-natural grasslands (in 2006 and 2007), that have been used as public pastures for cattle grazing for decades. The study grasslands are distributed throughout the Chernivtsi Region (47˚43'– 48˚41' N × 24˚55'– 27˚30' E) located in the south-west of Ukraine along the rivers Dniester, Prut and Siret (Fig 2). Pastures account approximately for 15% of the agricultural land area in the region. The study area generally experiences a temperate humid continental climate, with high precipitation amounts, which increase with increasing elevation (see below for details). Although the Chernivtsi Region is the smallest within Ukraine (Fig 2) by land area (8,097 km$^2$), it is highly diverse in environmental conditions. According to the

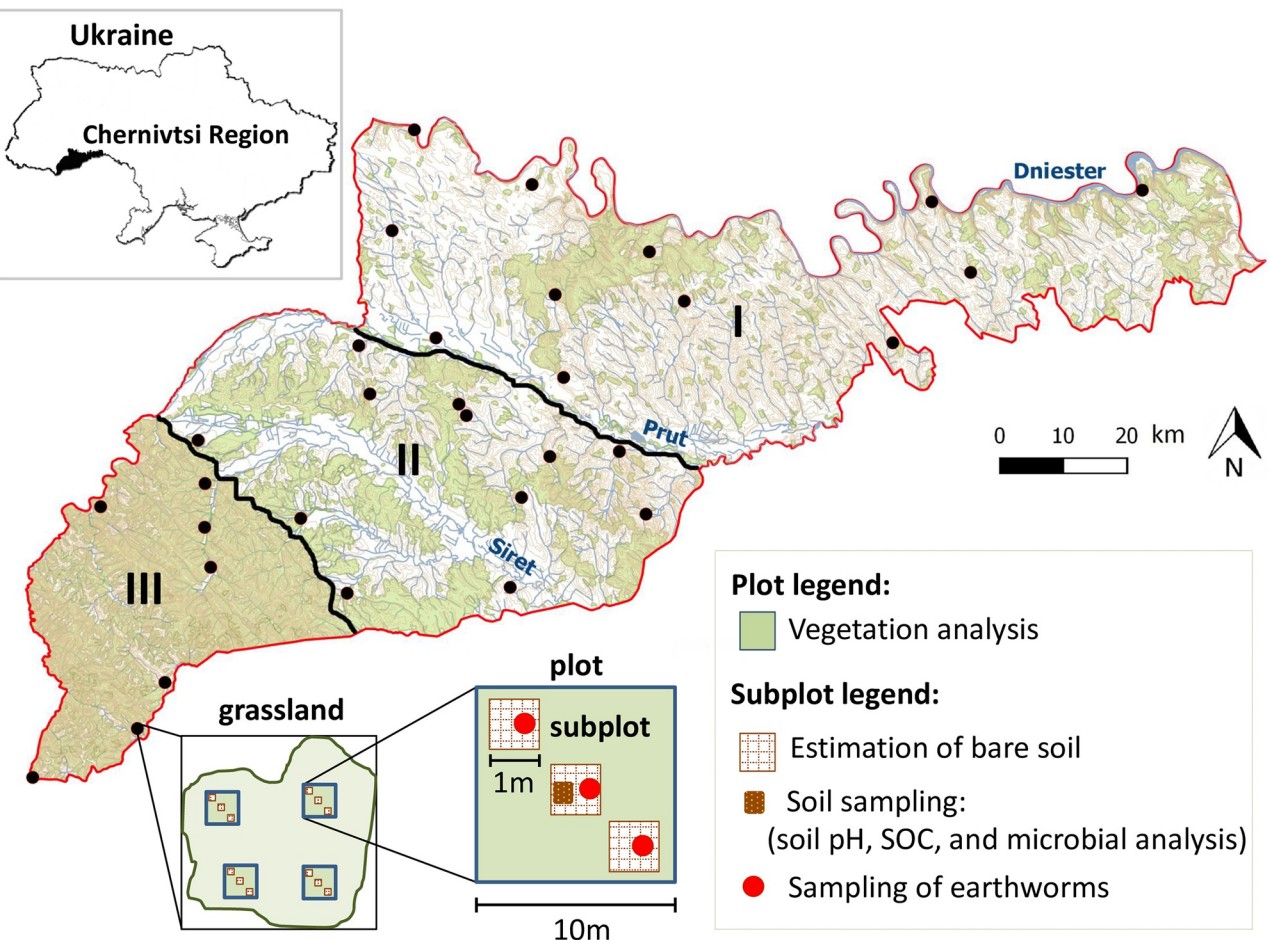

**Fig 2. Location of studied grassland sites (filled dots, n = 31) across the study area (Eastern Carpathians, Chernivtsi Region, Ukraine) and experimental design.** Physical-geographical zones: I plains (n = 12 grassland sites); II foothills (the Precarpathians) (n = 12 grassland sites); and III mountains (n = 7 grassland sites). Green areas show forest habitats; and white areas show unforested lands. The rivers Dniester, Prut, and Siret are major rivers of Chernivtsi Region. SOC, soil organic carbon. The map was created using QGIS 2.18.26 [102]. Four plots (10 m × 10 m) per grassland site and three subplots (1 m × 1 m) within each plot were selected.

specific characteristics of landscapes, climate, elevation, and age and type of rocks it can be divided into the three physical-geographical zones: *the plains*, *the foothills* (the Precarpathians) and the *mountains* (the Carpathian Mountains) (Fig 2) [50].

The plains of the Chernivtsi Region are located within the southwestern edge of the East European Plain. The landscape is mainly flat and densely dissected by river valleys and ravines. The plains represent ~49% of the territory of the region (Fig 2) and are located between the rivers Prut and Dniester (the Prut-Dniester interflow). The average elevation is 230 m a.s.l. with the highest elevation 515 m a.s.l. on hill formation in the center of the zone. The average annual rainfall is 575–660 mm, and the mean annual temperature is 8 ˚C with the averages of -5 ˚C in January and 20 ˚C in July. The soils are mostly Luvisols (pH = 5.2–6.5, humus = 1.5 –4%), Phaeozems (pH = 5.5–6.8, humus = 4–6%), and Chernozems (pH = 6.5–7, humus = 5 –12%) (S1 Fig in S1 Appendix). A high level of summer precipitation causes frequent soil erosions. The plains zone is characterized by deciduous forest vegetation (*Fagus sylvatica* L., *Quercus robur* L., and *Carpinus betulus* L.) [50]. In the west of the area there are meadow steppes

with *Festuca valesiaca* Schleich. ex Gaudin, *Poa angustifolia* L., *Carex humilis* Leyss., and *Brachipodium pinnatum* (L.) P. Beauv., each dominating in different patches [50].

The Precarpathians (the foothills) cover ~26% of the Chernivtsi Region and are located in the center of the region (Fig 2). The Precarpathians are formed as a part of the foredeep at the outer base of the Carpathian Mountains (i.e., part of the Outer Carpathian Depression). They are considered the foothills of the Ukrainian Carpathians and represent a transition from the plains to the mountains. The terrain is highly complex due to the predominance of the wavy hills, densely dissected by the rivers Prut and Siret. The average elevation is 350 m a.s.l. (rising up to 537 m a.s.l.); and the average annual rainfall is 575–780 mm. The mean annual temperature is 7 ˚C with the average temperatures in January -4.5˚C and in July 18 ˚C. The soils are mostly Retisols (pH = 4.3–4.5, humus = 1–3%) (S1 Fig in S1 Appendix). Luvisols soils also can be found over the entire foothills (S1 Fig in S1 Appendix). The vegetation mostly consists of broad-leaved forests (*Fagus sylvatica* L., *Carpinus betulus* L., *Acer pseudoplatanus* L.), mixed fir-spruce (*Abies alba* Mill.–*Picea abies* (L.) H. Karst.) and fir-beech (*Abies alba* Mill.–*Fagus sylvatica* L.) forests, and mixed forb – graminoid meadows [50].

The mountain zone covers ~25% of the Chernivtsi Region and is part of the Carpathian Mountains (i.e., the Outer Eastern Carpathians) with an average elevation of about 900 m a.s.l. (from ~600 to 1565 m a.s.l.). The climate is colder (mean annual temperature: 4.6 ˚C with the average temperature in January -7.4 ˚C and in July 15 ˚C) and excessively humid (annual rainfall: 700–1200 mm) in comparison with the other zones. The relief is a typical mountain relief with a variety of landscapes such as low-hill terrains covered by forests and secondary meadows; mountains with a medium elevation covered by forests; and subalpine mountains covered by subalpine meadows. Soils are mostly Cambisols (pH = 4.3–5.5, humus = 1–8%), and Retisols (S1 Fig in S1 Appendix). The prevailing vegetation closely follows the following elevation lines [50]: European beech (*Fagus sylvatica* L.) and beech-fir (*Fagus sylvatica* L.—*Abies alba* Mill.) forests at 800–950 m a.s.l.; fir-spruce (*Abies alba* Mill.—*Picea abies* (L.) H. Karst.) forests at 950–1100 m a.s.l.; spruce (*Picea abies*(L.) H. Karst.) forests at 1100–1400 m a.s.l.; shrublands at > 1400 m a.s.l.; and polonynas that are subalpine grassland landscapes in the Ukrainian part of the Eastern Carpathian Mountains, which developed above the upper forest limit within the elevation range of 1200 and 2000 m a.s.l. [51].

## Site selection

Of the 31 grasslands sites, 12 were selected within the plains, 12 within the foothills and 7 within the mountains. The unbalanced study design was not an issue in this study because we did not consider the effects of physical-geographical zones while instead we tested the effects of elevation as a continuous variable. All grasslands have been used as common grazing lands for cattle pasturing by private households, which typically have 2–3 livestock units per household (for calculations of livestock density in units per area see explanations below and S2 Table in S1 Appendix). Grazing season depends on the growing season of vegetation and varies for the three physical-geographical zones. The grazing season for the plains lasts nearly 210–220 days, for the foothills around 180 days and for mountain grasslands 120–150 days. Sampling was performed identically for each of the compared ecosystems during June–July in 2006 and 2007. The study was approved for the collection of plant, soil, and soil biota samples by the Scientific Council of Chernivtsi National University (protocol number 1/23.02.2006). The field studies did not involve endangered or protected species. The same grassland sites were sampled twice, i.e. one time per year. A handheld GPS-12 Garmin® (±15m accuracy) was used to identify the geographic coordinates and average elevation (m a.s.l.) for each study ecosystem. Four plots (10 m × 10 m) were selected within each of the 31 grassland sites (Fig 2).

Placement of plots was random but constrained by the edges and size of the field site. The average closest distance between two neighboring plots was 10 m and was chosen to minimize the potential for spatial autocorrelation influencing the results. The distance from plot to the edges of the grassland was 10 m to prevent edge effects. Within each plot, a transect was positioned diagonally through the plot. Three 1 m by 1 m subplots for estimation of bare soil, and earthworm sampling were randomly selected along the transect with a minimum distance to the next subplot of 1 m (Fig 2). Soil samples for microbial and chemical analysis were taken from one of the three subplots within each plot (Fig 2). Locations of plots and subplots within each grassland site differed across the two years.

## Cattle density and bare soil exposure

Cattle density was measured as the number of livestock units per hectare of grassland area (livestock units h$^{-1}$) (S2 Table in S1 Appendix). For this, the number, type and age of cattle were recorded for private households, which used the pastures during the grazing seasons of the two study years. We transformed the number of cattle to livestock units based on the widely used conversion factors for Europe (S2 Table in S1 Appendix) and used this cattle density measure as a proxy for the degree of grazing intensity of the study pastures. For each 1 m$^2$ subplot, the fraction of bare soil was visually estimated. The mean of the three subplots and then the four plots was taken to approximate the percent bare soil per grassland in each year. Cattle density and the fraction of bare soil were averaged across the two sampling years.

## Vegetation

Vegetation was recorded within each plot during the peak growing season for the different physical-geographical zones (June–July) in the two study years. All plant species were determined within each plot. We used the number of species as a measure of plant species richness per 100 m$^2$. Within each plot three 1 m × 1 m randomly selected subplots were used to estimate the relative cover of each species by vertically projecting canopy cover (%) for each species within each subplot. Averages were then taken across the three replicate subplots. All recorded species were classified into four functional groups: legumes (Fabaceae), grasses (Poaceae), other monocots (rushes and sedges), and forbs (other than legumes). The number of functional groups was used as the measure of plant functional diversity per 100 m$^2$. Further, species were classified as undesirable for grazing (S4 Table in S1 Appendix) [52–54] if they were known to reduce grazing efficiency, forage yield, palatability and quality, therefore contributing to lower forage and animal production of grassland ecosystems [55]. The group of undesirable species includes both unpalatable species as well as competitive weeds. Unpalatable plants are those containing toxic compounds poisonous to cattle (e.g., *Equisetum arvense*, *Ranunculus acris*, *Saponaria officinalis*, *Euphorbia* sp.), or, when eaten, may cause mechanical injuries because of a spiny covering or fine hairs (e.g., *Carduus crispus*) [53]. Competitive weeds (e.g., some coarse tall grasses and forbs) are not toxic to cattle and somewhat palatable (e.g., *Plantago* sp.), but can increase in density over time and outcompete desirable forage species. In a pasture, they reduce grazing efficiency of cattle by increasing search time for high-quality food. The number of undesirable species was quantified on the 100 m$^2$ plots. Data on plant species richness, functional diversity and richness of undesirable species were averaged across the four plots level for each year and further averaged across the two sampling years. Further, we measured the proportion of undesirable species as the ratio of their species number to the total plant species number. Data on canopy cover (%) for each species were averaged across the four plots within each year and scaled to unitless relative cover measures (C) ranging from 1 to 5 in accordance with Braun-Blanquet [56]: C = 1 for cover from 1 to 5%; C = 2 for

cover from >5 to 25%; C = 3 for cover from >25 to 50%; C = 4 for cover from >50 to 75%; C = 5 for cover >75%. Further, the relative cover measures for each species were averaged across the two sampling years.

## Soil sampling and analysis

Before soil sampling, vegetation and upper litter layer were removed in a small area. One soil sample per plot (for a total of four samples per grassland, Fig 2) was collected at 0–10 cm depth during the vegetation sampling campaign within each of the grassland sites using a soil corer with a diameter of 5 cm. The soil samples were immediately stored at -5 ˚C for microbial and chemical analysis. For the examination of organic carbon content and soil pH analysis the soil samples were air-dried, sieved (mesh width 2 mm) and homogenized. Soil organic carbon (%) was determined using a Tyurin's wet combustion technique, which is based on organic carbon oxidation by potassium dichromate (0.4 N) in acid solution ($K_2Cr_2O_4$: $H_2SO_4$ in a 1:1 ratio). The soil pH was determined by a standard glass electrode pH meter using a potassium chloride solution in a 1:2 ratio (soil: 0.1-N KCl).

## Density of soil biota

For the soil microbiological analysis, we counted cells of three microbial groups: heterotrophic bacteria, micromycetes, and actinomycetes. Cells were cultured on group-specific substrates under controlled temperature conditions: heterotrophic bacteria were cultured on meat-peptone agar between 28 and 30˚C, micromycetes were cultured on modified Czapek-Dox substrate with streptomycin at 20 to 25˚C, and actinomycetes were cultured on starch-ammonium agar at 28 to 30˚C. The total number of cells of all three groups was used as abundance measure for the soil microbial community (cells $\times 10^8$ $g^{-1}$ dry soil). We took averages across the four plots (Fig 2) to approximate the abundance per grassland for each year. Further, the abundance data were averaged across the two sampling years. To sample earthworms we used a standard Quantitative Hand Sorting method. For this, 30×30 $cm^2$ soil blocks with a depth of 15 cm were excavated from each of three subplots in each of four plots (leading to 12 samples per grassland, Fig 2), and earthworms were immediately separated manually in the field and sampled into empty vials. At the same day specimens were counted in the lab, their fresh weight was determined, then they were oven-dried and their dry weight was determined. Earthworm dry weight data were calibrated to an area of 1 $m^2$. Averages were taken across the four plots to calculate earthworm biomass per grassland (g $m^{-2}$) for each year and further averaged across the two sampling years.

## Data analysis

All analyses were performed in R version 3.4.3 [57]. We applied structural equation modelling (SEM) using the package 'lavaan' [58] in R as an exploratory approach for assessing direct and indirect simultaneous effects of grazing intensity on plant species richness, functional diversity and proportion of undesirable species along the elevation gradient. We summarized the data at the site level (i.e. mean for the entire grassland) and further across the two sampling years to create a single-level data set for each study grassland because we seek to understand the elevational variation rather than temporal or within-site variation. The traditional SEM, as used in the current study, is an appropriate tool for analysing such single-level data sets, because it assumes that there is no underlying structure to the data [49], such as effects of plots within sites or random effects of the sampling year. The direct grazing effects in our study (Fig 1, *path 1*) assume that grazing intensity impacts the plant variables independently of grazing-induced variations in soil biotic and abiotic properties. The indirect effects of grazing infer that grazing

 

intensity impacts the plant variables through the changes in bare soil exposure (Fig 1, *pathway 2→3*), soil chemical properties (Fig 1, *pathway 9→8* and *pathway 2→10→8*), or soil biocommunity variables (Fig 1, *pathway 4→5*; *pathway 2→6→5*; *pathway 4→7→8*). The direct elevation effects (Fig 1, *path 11*) assume that the impact of elevation on the plant variables is independent of the elevation-induced alterations of grazing intensity and soil properties. The indirect effects of elevation in our study assumes that the plant variables vary across the elevation gradient through the changes in grazing pressure (Fig 1, *pathway 12→1*), soil conditions (Fig 1, *pathway 13→8*; *pathway 15→3*) or the follow-up alterations in soil biocommunities (Fig 1, *pathway 13→14→5*; *pathway 15→6→5*). SEM allows for the inference of such indirect effects from observational data in complex systems by analysing the covariance structure of multiple variables [49]. At first, we constructed a hypothetical model (Fig 1) to allow for direct and indirect effects of grazing intensity, elevation, soil properties, and soil biota on plant response variables (see S1 Table in S1 Appendix for ecologically meaningful relationships that we hypothesized based on the literature). Although soil biota may influence bare soil exposure, here we assumed prevalence of grazing-induced bare soil impacts on the soil community (Fig 1, *path 6*) because we expected the direct effect of cattle trampling to be the most severe on bare soil exposure (Fig 1, *path 2*) [25]. Further, we tested the hypothetical model (Fig 1) with our sampled data (Fig 3, S6 Table in S1 Appendix). Plant species richness and plant functional diversity were log-transformed; and cattle density was square-root transformed to meet the assumptions of normality and homoscedasticity. The Pearson correlations between all predictor variables were lower 0.70 (S10 Table in S1 Appendix) and therefore may well be included in our multivariate analyses, in accordance with Tabachnick and Fidell [59]. We used a Chi-square ($\chi^2$) test with maximum likelihood ratio, Root Mean Square Error (RMSEA), Comparative Fit Index (CFI) and Tucker–Lewis Non-Normed Fit Index (NNFI) and Standardized Root Mean Square Residual (SRMR) as goodness of fit tests to assess the validity of the SEM model [49]. The individual effects included in the model were evaluated for significance (the *P*-value is lower than α = 0.05), and standardized SEM regression coefficients were used as a quantitative measure of the strength of these effects (S6 Table in S1 Appendix, Fig 2). We also tested the significance of the mediation effects, i.e. all indirect paths (effects among two variables mediated by other variables) were also evaluated for significance (S7 Table in S1 Appendix). To assess whether the grazing-intensity effects on plant community varied across the elevation gradient, we tested for an interactive effect of grazing and elevation (Fig 1, *path 16*) on each of the response variable, i.e. on plant species richness, functional diversity, and the proportion of undesirable species. The interactive effects of grazing and elevation (Fig 1, *path 16*) were not deemed significant in the original SEM model (S2 Fig in S1 Appendix), and they were removed in the final model (Fig 3) to achieve adequate fit statistics. The final SEM model (Fig 3, S6 Table in S1 Appendix) was well supported by the data ($\chi^2$ = 10.2, df = 9, P = 0.33, RMSEA = 0.07, $P_{RMSEA}$ = 0.39; CFI = 0.99; NNFI = 0.96; SRMR = 0.05), and an addition of any other paths did not improve the model, suggesting that all important relationships were specified.

To investigate plant community composition we used nonmetric multidimensional scaling (NMDS) based on Bray–Curtis matrices using the 'metaMDS' function in the *vegan* package in R [60] with a maximum of 100 random starts. We plotted differences in community composition using an NMDS plot and fitted grazing intensity and elevation post hoc using the 'envfit' function and the 'ordisurf' function, respectively. To test the effects of elevation, cattle density, soil properties, and soil biota on plant community composition we performed a PERMANOVA test on Bray-Curtis matrices with 1000 permutations using the 'adonis' function in the *vegan* package in R [60].

Given that the study sites are located in three different physical-geographical zones (Fig 1), which are also associated with elevation and grazing intensity, we tested whether the spatial

 

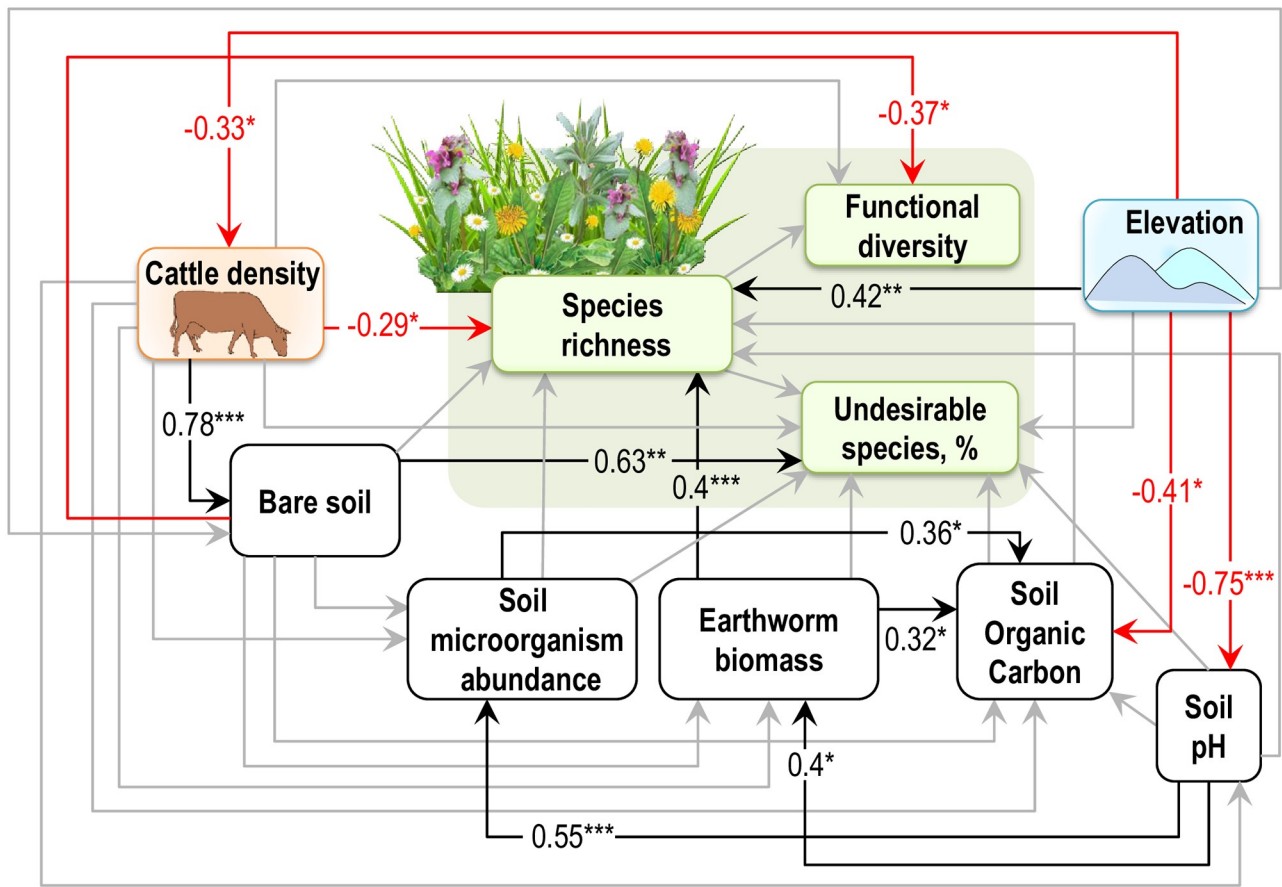

**Fig 3. SEM showing the direct and indirect pathways through which grazing intensity and environmental conditions affected plant diversity and composition.** Models were well supported by our data ($\chi^2$ = 10.2, df = 9, P = 0.33, RMSEA = 0.07, $P_{RMSEA}$ = 0.39; CFI = 0.99; NNFI = 0.96; SRMR = 0.05). Numbers associated with each arrow are standardized path coefficients with the following levels of significance: $^{*}$P $\leq$ 0.05; $^{**}$P $\leq$ 0.01; $^{***}$P $\leq$ 0.001. Red and black paths represent significant (i.e., P $\leq$ 0.05) negative and positive path coefficients, respectively, and grey paths were not significant. RMSEA: Root Mean Square Error of Approximation; CFI, Comparative Fit Index; NNFI: Tucker–Lewis Non-Normed Fit Index; SRMR: Standardized Root Mean Square Residual. See S6 Tables in S1 Appendix for more details.

autocorrelation among study sites affected our results using Moran's I statistic of residuals for each study variable. The residuals were extracted ('resid' function in R) from linear models ('lm' function in R) with the same combinations of predictor variables for each respective response variable as tested in the SEM (Fig 1, S6 Table in S1 Appendix), in variance analysis (for species richness of each functional group of plants, S8 Table in S1 Appendix), and in PER-MANOVA (for plant community composition, S9 Table in S1 Appendix). The spatial matrix of weights for Moran's I test was calculated ('dist' function in R) as the inverse distance matrix (Euclidean distances between pairs of sites) based on longitude and latitude data for each study site. Moran's I statistic was calculated for model residuals of each response variable (S11 Table in S1 Appendix) using the 'Moran.I' function in the *ape* package in R [61]. The analysis of Moran's I coefficient revealed no significant autocorrelation among residuals for any of the response variables (as all P>0.05, S11 Table in S1 Appendix) except soil pH, for which Moran's I coefficient was close to zero. Overall, this suggests that spatial autocorrelation among study sites did not affect our results.

## Results

In the 31 grasslands (Fig 2) we sampled a total of 175 plant species: 13 legumes (Fabaceae), 18 grasses (Poaceae), 8 other monocots (i.e., rushes and sedges), and 136 non-legume forbs. Plant species richness varied from 7 to 51 species per grassland (S3 Table in S1 Appendix). The list of species and summary statistics of the richness derived from the plant community data for each of the physical-geographical zones are given in S3 and S4 Tables in S1 Appendix. In total, we classified 86 species as undesirable for grazing (thereof 48 species in the plains, 60 species in the foothills, and 62 species in the mountains, see S3 Table in S1 Appendix), which varied from 5 to 28.5 species per grassland (13.9±1.08; mean±S.E.). Cattle density of the study grasslands varied from low (0.01 livestock units $h^{-1}$) to high (1.38 livestock units $h^{-1}$), and elevation varied from 399 to 1323 m a.s.l. across the grassland sites (S5 Table in S1 Appendix).

### Drivers of plant species richness

Overall, the sampled plant communities were rich in species number (27.5±1.97, mean±S.E.), reaching a maximum of 51 species per grassland site (S3 Table in S1 Appendix). We found that cattle density significantly reduced plant species richness (Fig 3). Disentangling direct and indirect effects using SEM (S7 Table in S1 Appendix) revealed only direct impacts of cattle density on species richness, while none of the indirect effects (i.e., mediated via bare soil exposure, soil properties, or soil decomposers) were significant (Fig 3, S7 Table in S1 Appendix). In contrast, elevation directly and indirectly influenced plant species richness in our grassland sites. The direct effect was positive while the indirect effect was negative, resulting in a weaker positive overall effect. The indirect negative effect of elevation was mediated via soil pH and earthworm biomass (S7 Table in S1 Appendix). Specifically, a decrease in soil pH with increasing elevation reduced earthworm biomass, which in turn resulted in lower plant species richness (S7 Table in S1 Appendix).

Cattle density decreased with increasing elevation in our grassland sites (Fig 3), however, grazing-mediated indirect effects of elevation on plant species richness were not significant (S7 Table in S1 Appendix). Furthermore, effects of cattle density on species richness did not vary across the elevation gradient, as the interactive effects of grazing and elevation on plant species richness were not significant (S2 Fig in S1 Appendix). In general, at our grassland sites the overall impact (direct and indirect effects) of cattle density on plant species richness was stronger than that of elevation (S7 Table in S1 Appendix).

### Drivers of functional diversity and community composition

All significant effects of cattle density on plant functional diversity and on percent of undesirable species in plant community were indirect and mediated by bare soil exposure (S7 Table in S1 Appendix). Specifically, greater cattle density led to an increased fraction of bare soil, which in turn decreased plant functional diversity (Fig 3). In contrast, grazing-induced bare soil increased percentage of undesirable species (Fig 3), therefore resulting in positive overall impact of grazing intensity on the proportion of undesirable species in a plant community. Cattle density reduced species number of legumes. Species number of non-legume forbs and of rushes and sedges increased with increased fraction of bare soil under greater grazing intensity (S8 Table in S1 Appendix). In contrast, species number of grasses was not influenced by cattle density in our grassland sites.

The SEM model indicated no significant overall effects of elevation on functional diversity or on the proportion of undesirable species in a plant community (S7 Table in S1 Appendix). When testing the effect on each functional group separately in linear models, we found a positive effect of elevation on the species number of rushes and sedges (S8 Table in S1 Appendix).

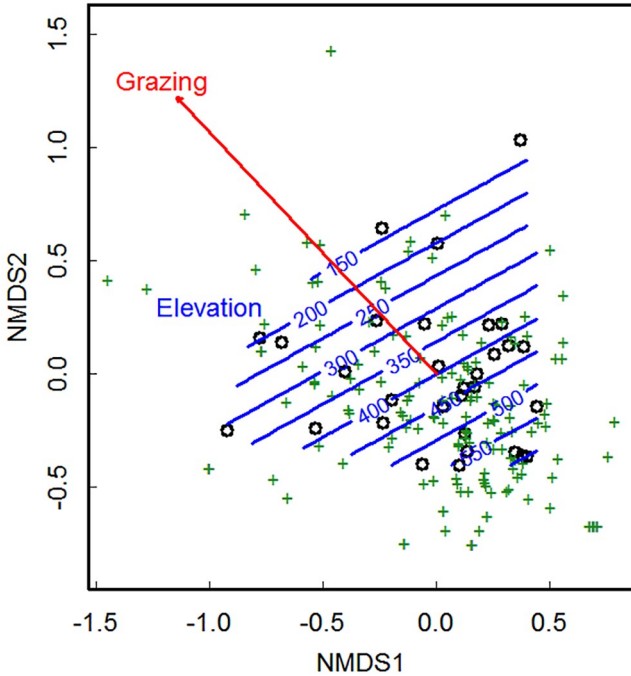

**Fig 4. Nonmetric multidimensional scaling (NMDS) plot showing the composition of plant communities in grasslands and the effects of grazing and elevation on plant community composition.** Symbols in green show species (n = 175), symbols in black show grasslands (n = 31). Grazing intensity and elevation significantly affected community composition (see S9 Table in S1 Appendix). Grazing intensity is shown by a red arrow drawn by fitting cattle density post hoc. Elevation gradient is illustrated by the blue contour lines. Stress = 0.16.

There were no significant effects of elevation on species number of legumes, non-legume forbs, or grasses (S8 Table in S1 Appendix). The interactive effects of grazing and elevation on plant functional diversity and on proportion of undesirable species were not significant (S2 Fig in S1 Appendix).

We found no significant effects of species richness on functional diversity and on the proportion of undesirable species (Fig 3). However, species number of legumes increased with both, community species richness and functional diversity. Species numbers of both grasses and other monocots (i.e., rushes and sedges) were significantly higher in more functionally diverse plant communities but not affected by community species richness. In contrast, species number of non-legume forbs was positively affected by community species richness but negatively affected by functional diversity (S8 Table in S1 Appendix).

Plant community composition was assessed based on individual species' canopy cover. The changes in community composition are shown in the NMDS plot (Fig 4). Both changes in elevation and grazing intensity significantly affected plant community composition (elevation: $F_{1,30}$ = 1.96, $R^2$ = 0.06, P = 0.005; grazing: $F_{1,30}$ = 1.63, $R^2$ = 0.05, P = 0.03), while the interactive effects of grazing and elevation as well as soil parameters were not significant (S9 Table in S1 Appendix).

## Discussion

### Effects of grazing

In our study grazing affected plant species richness, functional diversity, the proportion of undesirable species (S7 Table in S1 Appendix), and plant community composition (Fig 4,

S9 Table in S1 Appendix). However, the underlying mechanisms of these effects differed. Increasing grazing intensity reduced plant species richness and this effect was driven mainly by a direct link between cattle density and plant species richness (Fig 3), while none of indirect effects here were significant (S7 Table in S1 Appendix). These findings are in agreement with Socher et al. [62] who also found that effects of grazing on plant species richness were governed by direct effects, in contrast to other land-use types, e.g. fertilization. This direct effect can likely be explained by the mechanical destruction of vegetation, overgrazing, and chemical and biological impact of faeces and urine depositions [25] (S1 Table in S1 Appendix), but possibly also by an altered colonization of species from regional species pools via the removal of seeds and reproductive structures of plants by grazing [63]. Diet preference of livestock for certain species and selective grazing [64, 65] is also one of the major factors shaping plant community composition along a grazing intensity gradient [66] because the survival of palatable species is hampered by both grazing pressure and competitive pressure from unpalatable species [66].

In contrast to species richness, the effects of grazing on functional diversity and on the proportion of undesirable species were mediated via bare soil exposure (Fig 3). Specifically, we found a reduction in functional diversity and an increase in the proportion of undesirable species along the increased fraction of the grazing-induced bare soil (Fig 3, S7 Table in S1 Appendix). The observed reduction in functional diversity was a result of the loss of legume species from the plant community under increasing grazing intensity (S8 Table in S1 Appendix). Most of legume species observed in our study are palatable to livestock (S4 Table in S1 Appendix), and may perhaps be favored among the other plant groups for grazing. Previous studies show that palatable legumes have generally lower persistence under grazing in contrast to other functional groups [66, 67], due to the preferential grazing of forage legumes by ruminants [68]. Furthermore, high inputs of nitrogen in the soil, delivered with livestock urine [25], have been previously found to suppress legumes [69]. Instead, the less palatable and more competitive species might further benefit from the grazing pressure [66], which may create favorable conditions for their seed germination [26, 70]. Our results are aligned with this prediction as we found that competitive undesirable weed species, which are predominantly rushes, sedges and some non-legume forbs (S4 Table in S1 Appendix), increased in species number with increased fraction of cattle-induced bare soil (S8 Table in S1 Appendix). This observed pattern confirms findings of other studies. For example, Tardella [35] found that the presence of bare soil, linked to lower soil depth, increased the number of stress-tolerant and ruderal plants, which are generally undesirable for grazing and such habitats further nurse competitive weeds. Similarly, Symstad [71] showed that an increased fraction of bare soil promotes the invasion success of highly competitive species. This general finding of a shift in plant community composition towards undesirable weed species under grazing can be attributed to several possible mechanisms. Compacted by cattle trampling bare soil reduces nutrient and water availability to plants [28, 33] and restricts the ability of plant seedlings to penetrate the soil and to emerge [25]. This may lead to a shift in community composition by favouring species with high competitive abilities for resource acquisition. On the other hand, an increased proportion of bare soil with increasing cattle density might lead to more pronounced spatial heterogeneity due to increased vegetation patchiness [27]. Such open canopy gaps then filter for adapted colonizer species [72], for example short-lived forbs [71, 72]. Thus, the tendency of undesirable species to increase in grazing-induced bare soil patches in our study may relate to both, their competitive ability for resources and to their ability of colonizing bare soil [72]. Ultimately, these mechanisms may explain the alterations in plant community composition under the intensification of grazing pressure, as shown in our study (Fig 4, S9 Table in S1 Appendix) and by previous findings [66].

We found no significant effects of cattle density or of bare soil exposure on soil pH and organic matter content. This is contrary to other findings reporting that cattle density, linked to increased cattle manure depositions, can relax soil acidification processes and to maintain soil pH and soil organic matter in optimal ranges [73]. Furthermore, neither cattle density nor bare soil in our study impacted the densities of soil microorganisms and earthworms, which is in contrast to earlier studies, reporting strong effects of grazing intensification and bare soil exposure on soil microbial biomass [74, 75], microbial activity [76], and earthworm abundance and biomass [31]. Thus, our results may imply that the range of soil compaction exerted by cattle on the studied grasslands was not strong enough to affect the abundance of earthworms and the soil microbial community [76–78].

## Effects of elevation

We found strong effects of elevation on plant community composition (Fig 4, S9 Table in S1 Appendix) and on plant species richness in our grassland sites (Fig 3, S7 Table in S1 Appendix). Conversely, we found no significant effects of elevation on functional diversity of the plant community or on the proportion of undesirable species (S7 Table in S1 Appendix). The positive effect of elevation on species number of rushes and sedges as observed in our study (S8 Table in S1 Appendix) might be attributed to the increased humidity with increasing elevation [40], as rushes and sedges are known to dominate other species in increased soil moisture conditions [79]. Elevation was found to impact plant diversity and community composition in previous studies [45, 80, 81], mainly because of the correlation with climatic factors [45, 81–83], for example due to higher rainfall at higher elevation [40]. Additionally, higher topographic variability with increasing elevation across our study area results in greater habitat diversity [42] and thus may allow plants to find suitable habitats within small distances, therefore potentially leading to higher species richness. Furthermore, studies show that competitive interactions in plant community decrease at higher elevations [84], thus the increase in plant species richness along elevation gradient in our study may be partially due to the reduced dominance of competitive species. The recent and ongoing proliferations of plant species upward in elevation due to global climatic changes [85] may also contribute to the observed shifts in community composition and increased local species richness with increasing elevation. Our results also show that the effect of elevation on plant species richness was partially negative and mediated by reduced biomass of earthworms as a result of increased soil acidification with increasing elevation (Fig 3, S7 Table in S1 Appendix). Decreased abundance of soil microorganisms and subsequent reduction in soil organic carbon also resulted from increased soil acidification along elevation gradient (Fig 3, S7 Table in S1 Appendix). Our results agree with the previous records showing that soil pH is one of the primary determinants of plant and soil-decomposer communities along elevation gradient in mountains [83]. The observed decrease in soil pH with increasing elevation in our study could be linked to higher soil leaching processes in more humid and lower temperature conditions with increasing elevation [40]. Furthermore, within our study area, the change in climatic conditions with elevation and topography determines the differences in the soil types (S1 Fig in S1 Appendix), which further result in variation of soil properties along the elevation gradient, including soil pH [40]. Increased soil acidity was found to suppress decomposition, burrowing, casting and mixing activities of earthworms with cascading consequences for growth [31] and competitive interactions among plant species [32]. Such differences in the soil types and soil abiotic and biotic properties may also explain the alterations in plant community composition along elevation gradient observed in our study (Fig 4, S9 Table in S1 Appendix).

## Interrelations and differences between the effects of elevation and grazing on plant community

Elevation had a weak impact on cattle density in our grassland site (Fig 3), leading to nonsignificant mediation of elevation effects on plant species richness, functional diversity, and on proportion of undesirable species through grazing (S7 Table in S1 Appendix). Furthermore, we found that the interactive effects of grazing and elevation on plant diversity and composition (S2 Fig in S1 Appendix) were not significant. These results indicate that the effects of grazing intensity on plant community in our grassland sites did not change across the elevation gradient. This is in contrast to some previous evidence that land use and other anthropogenic influences in mountains impact the elevational patterns of plant diversity and composition [36, 86]. However, the elevation-dependency of the effects of grazing on vegetation may vary with different grazing management strategies. For instance, Speed *et al.* [36] found a positive relationship between grazing-induced change in species richness and elevation on grasslands where grazing was reduced. However, where grazing was maintained or increased, changes in species richness with grazing intensification did not vary along the elevational gradient.

Compared to grazing, elevation expressed a relatively stronger direct effect on plant species richness in our grassland sites (Fig 3). This is in agreement for example with Grace and Pugesek [80] and Báldi *et al.* [13] reporting that abiotic conditions exerted a relatively strong control over species richness, with grazing playing a less important role. However, the opposing direct and indirect effects of elevation on plant species richness in our grassland sites led to weaker positive overall effect. Furthermore, the same signs of the direct and indirect effects of grazing on plant species richness led to strong negative overall effect. As a result, an overall impact of cattle density on plant species richness was stronger than that of elevation in our grassland sites (S7 Table in S1 Appendix). Importantly, the indirect effects of grazing and elevation operated via different independent pathways, i.e. grazing affected plant community through bare soil exposure while elevation–via altered soil pH and soil decomposers. Our results support previous findings suggesting that land use and environmental factors may alter biocommunity via dissimilar mechanisms [87]. These results may be important in the context of global change, because land use and environmental factors are of the most dominant global change drivers.

## Potential explanations of inconsistencies in previous research

The inconsistencies in existing evidence on plant diversity responses to grazing may be partially explained when simultaneously considering different specific mechanisms potentially underlying grazing–biodiversity relationships along elevation gradient. Disentangling and simultaneously assessing different plant diversity metrics might potentially explain the different outcomes. For instance, in our study we found negative effects of grazing intensity on plant species richness and on plant functional diversity but positive effects on the proportion of undesirable plant species. These results may be attributed to differences in species' grazing-tolerance traits between study sites. However, studies confirm that contrary responses to grazing occur at high rates also at species level [88, 89] and may arise from differences in the abiotic or biotic contexts between sites, for example variations in soil quality [35, 89], increase in other competitive species (e.g., of undesirable species in our study) [90–92], and varying climate conditions [45, 81, 82] for instance along elevation gradients [45, 80, 81]. In the current study, along with grazing effects, we considered soil abiotic and biotic properties and between-site variations in elevation and found that some of the observed grazing effects on the plant community were directly linked to cattle density (i.e., the effects on species richness), while others (i.e., the effects on functional diversity and the fraction of undesirable species) were

mediated by bare soil exposure. Furthermore, elevation had both a positive direct effect on plant species richness and a negative indirect effect, which was mediated via altered soil pH and earthworm density. Our results suggest that discounting such underlying mechanisms when assessing grazing-biodiversity relations may lead to overlooked or underestimated effects of grazing on plant community, because a single mechanism may produce multiple patterns (e.g. negative effects on plant functional diversity but positive effects on undesirable species, Fig 3), while multiple mechanisms may lead to trade-offs among different effects (e.g. opposing positive direct and negative indirect effects of elevation on plant species richness result in overall weaker positive effect, S7 Table in S1 Appendix).

## Consequences for ecosystem resilience, functions and services

The variation in plant diversity and composition caused by grazing along the environmental gradients reported here may have consequences for multitrophic biodiversity [18, 93] and ecosystem functioning and services mediated by land use [3, 4, 6, 7, 22]. Cattle production itself is a valuable service but if other ecosystem services of the grasslands are to be considered, cattle grazing affect them. Our results show that high livestock stocking rates can directly induce changes in plant species richness, therefore likely influencing resource quality [3] and the availability and heterogeneity of resources to a wide range of organisms with possible cascading effects on biodiversity across trophic levels [18, 22, 93–95] and related ecosystem processes [4], such as primary productivity [3, 48], herbivory, decomposition, and predation [32, 96, 97]. Furthermore, the physical impact of the heavy livestock on soil resulted in bare soil exposure in our grassland sites with some of the most consistent outcomes for plant community composition being reduced plant functional diversity and increased proportion of undesirable species. This is likely to reduce grazing efficiency, forage yield and quality, therefore contributing to lower forage and animal production of grassland ecosystems [98]. Our results on a grazing-induced increase of bare soil entail important implications for the management of ecosystem functions and services, since previous studies show that bare soil exposure increases soil erodibility [38], leaching and plant invasion processes [71, 99], loss of nitrogen and pollution of the atmosphere with $N_2O$ during denitrification of waterlogged areas [25], and alters ecosystem watershed function [20].

Most of the Carpathian's unique biodiversity is dependent on semi-natural grasslands [100], which in turn require regular grass removal, such as via grazing, in order to survive [2]. However, as our results indicate, the benefits of grazing depend on the grazing intensity. As shown by our data, while low levels of grazing create high-diversity plant communities, high stocking rates reduce plant species and functional diversity and alter plant composition. On the other hand, farmland abandonments all over Europe, ongoing for last three decades [100] lead to the replacement of grasslands with successional shrublands, thus reducing ecosystem biodiversity and functioning [6]. Furthermore, our results indicate that grazing effects on soil and related ecosystem functions and services in semi-natural grasslands are likely more pronounced along the elevation gradient. The reduced soil organic carbon with increasing elevation shown by our data (Fig 3) might deteriorate the soil's ability to resist trampling pressure by cattle in our grassland sites [25] and to decrease grassland productivity [48]. We also found a lower density of earthworms and soil microorganisms with elevation, potentially impacting nutrient cycling, resource availability for plants, and plant productivity of grasslands [31].

## Conclusions

The current study provides data from a largely under-represented region in the scientific literature, the Eastern Carpathians in Ukraine. If we extrapolate our results to the entire region,

they suggest implications for land-use management strategies of the Carpathian semi-natural grasslands, where preserving biodiversity is crucial for the maintenance of ecosystem functioning and provisioning of ecosystem services [4, 13, 17, 20]. Our main findings are that plant diversity and composition are controlled by the complex interplay among grazing intensity and environmental conditions along the elevation gradient. Both, grazing and elevation affected plant community directly and indirectly via altered soil properties and soil biota. Thus, the assessment of land-use effects on grassland biodiversity should encompass simultaneous observations of multivariate parameters and links among them. Maintaining low levels of grazing in the semi-natural grasslands and taking into account the ecosystem characteristics driven by elevation and topography are therefore essential for maintaining species-rich and functionally diverse plant communities. Diversifying livestock [101] and seeding of diverse plant mixtures containing legumes and other desirable species [3] may be considered as additional strategies to promote biodiversity and multifunctionality in semi-natural grasslands for sustainable land-use intensification.

## Supporting information

**S1 Appendix.**
(DOCX)

## Acknowledgments

We are grateful to Vasyl Budzhak (Department of Botany, Forestry and Landscape Architecture, Chernivtsi National University) for help with species identification, to Christoph Scherber (University of Muenster) for discussions on the conceptual model, and to all colleagues from the Department of Ecology and Biomonitoring (Chernivtsi National University) and Theoretical Ecology Group (Freie Universität Berlin) for discussions. We thank Stepan S. Kostyshyn for management support and for facilities provided for field work and laboratory analyses. Further, we acknowledge the technicians and student helpers for their work.

## Author Contributions

**Conceptualization:** Oksana Y. Buzhdygan, Jana S. Petermann.

**Data curation:** Oksana Y. Buzhdygan.

**Formal analysis:** Oksana Y. Buzhdygan.

**Funding acquisition:** Oksana Y. Buzhdygan, Svitlana S. Rudenko, Jana S. Petermann.

**Investigation:** Oksana Y. Buzhdygan.

**Methodology:** Oksana Y. Buzhdygan.

**Resources:** Britta Tietjen, Svitlana S. Rudenko.

**Supervision:** Britta Tietjen, Svitlana S. Rudenko, Jana S. Petermann.

**Visualization:** Oksana Y. Buzhdygan, Volodymyr A. Nikorych.

**Writing – original draft:** Oksana Y. Buzhdygan.

**Writing – review & editing:** Oksana Y. Buzhdygan, Britta Tietjen, Svitlana S. Rudenko, Volodymyr A. Nikorych, Jana S. Petermann.

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
