## [Decision Letter · Decision Letter 0]

18 Jun 2020

PONE-D-20-07402

Direct and indirect effects of land-use intensity on plant communities across elevation in semi-natural grasslands

PLOS ONE

Dear Dr. Buzhdygan,

Thank you for submitting your manuscript to PLOS ONE. After careful consideration, we feel that it has merit but does not fully meet PLOS ONE’s publication criteria as it currently stands. Therefore, we invite you to submit a revised version of the manuscript that addresses the points raised during the review process.

We appreciate that your study shows the whole complexity of the issue rather than oversimplifying it, and that it refers to a regions largely underrepresented in scientific literature. However, the reviewers have a couple of concerns, some of them referring to data analysis: In a revised version, you may wish to describe / discuss more explicitly how spatial autocorrelation among sampling plots was handled. In addition, elevation, grazing intensity and soil potentially affect community composition, but they are correlated themselves. I think that SEM was used to disentangle their effects, but according to the reviews, this should be discussed more explicitly as well.

According to one of the reviewers, it becomes not clear if information included in Table S1 and Fig. 1 might already be a result of your study. I think figure and table just underpin and detail your main hypotheses about direct and indirect effects, and are therefore rather useful for understanding the rationale and the approach of your study. I suggest, however, referring to this issue more explicitly in the Introduction, e.g. that Fig. 1 shows the conceptual graph underlying your study in general and the SEM in particular. Furthermore, it should be deduced in the Introduction which hypothesis was intended to test using NMDS.

There are several further, rather minor comments provided by the reviewers, that should be carefully considered in a revised version of your manuscript.

We look forward to receiving your revised manuscript.

Kind regards,

Harald Auge

Academic Editor

PLOS ONE

Journal Requirements:

3. We note that Figure 2 and FigS1 in your submission contain map images which may be copyrighted. All PLOS content is published under the Creative Commons Attribution License (CC BY 4.0), which means that the manuscript, images, and Supporting Information files will be freely available online, and any third party is permitted to access, download, copy, distribute, and use these materials in any way, even commercially, with proper attribution. For these reasons, we cannot publish previously copyrighted maps or satellite images created using proprietary data, such as Google software (Google Maps, Street View, and Earth). For more information, see our copyright guidelines: http://journals.plos.org/plosone/s/licenses-and-copyright.

3.1.    You may seek permission from the original copyright holder of Figure 2 and FigS1 to publish the content specifically under the CC BY 4.0 license.

3.2.    If you are unable to obtain permission from the original copyright holder to publish these figures under the CC BY 4.0 license or if the copyright holder’s requirements are incompatible with the CC BY 4.0 license, please either i) remove the figure or ii) supply a replacement figure that complies with the CC BY 4.0 license. Please check copyright information on all replacement figures and update the figure caption with source information. If applicable, please specify in the figure caption text when a figure is similar but not identical to the original image and is therefore for illustrative purposes only.

Reviewers' comments:

Reviewer's Responses to Questions

**Comments to the Author**

1. Is the manuscript technically sound, and do the data support the conclusions?

Reviewer #1: Partly

Reviewer #2: Partly

2. Has the statistical analysis been performed appropriately and rigorously? 

Reviewer #1: Yes

Reviewer #2: No

3. Have the authors made all data underlying the findings in their manuscript fully available?

Reviewer #1: Yes

Reviewer #2: Yes

4. Is the manuscript presented in an intelligible fashion and written in standard English?

Reviewer #1: Yes

Reviewer #2: Yes

5. Review Comments to the Author

Reviewer #1: Paper review PONE-D-20-07402

Title: Direct and indirect effects of land-use intensity on plant communities across elevation in semi-natural grasslands

General comments:

This is a well-written manuscript about an interesting topic: Direct and indirect effects of land-use intensity on plant communities across an elevational gradient. The studied plant communities were semi-natural grasslands in the Eastern Carpathians in Ukraine. This region is in fact largely under-represented in the scientific literature, as the authors state in their conclusion. Further, I like the combination of vegetation data with data from other organisms, e.g. earthworms. The dataset is therefore promising, but I have some concerns about the analysis and the presentation of the data.

My largest concern is about the relation between the elevation and the grazing intensity. The authors show in the results that grazing intensity is negatively related to elevation, which is quite logic. Further, they quote that grazing intensity has large direct effects on the vegetation independent from elevation. I cannot see, in the current version of the manuscript, how the applied methods can disentangle this interrelationship. This leads me to another concern: I am not sure if the data presentation is optimal. While some tables in the supporting information seem to be relevant for understanding the argumentation of the data analyses (e.g. table S5 and S6), I do not see the benefit of figure 4, which shows the relationship between elevation and grazing intensity. I think more emphasize has to be laid on how this overlay of the gradients (elevation, grazing intensity and soil types as shown by Fig. S1) was disentangled.

Another question I have is concerned with the introduction: For me it is quite unusual to cite tables of the supporting information and to include a figure in the introduction. Does this fit to the journal guideline of PlosOne? Additionally, which methods were used to come up with figure 1, is it already a product of this study? Then I would handle it as such.

Overall, I recommend major revisions of this article.

Minor comments:

Abstract:

Line 24: most vulnerable compared to what?

Line 26: “may depend on environmental conditions and indirect effects are rarely considered” is very vague. Could you specify what you mean?

Line 29-30: The cattle grazing took place for two years? I guess you mean your sampling, you have to change the structure of the sentence.

Introduction:

Fig. 1: Why has the elevation no influence on the soil community? Could the soil community not influence the bare soil? When was this figure produced, was it for this paper? Maybe it is already a result of the study?

Line 133: I recommend to cite your map (figure 2) in the section about the study area.

Line 133-137: I would recommend formulating more specific research questions or hypotheses.

Methods:

Line 142: “for decades” to the end of the sentence.

Line 146: “significant amount of precipitation” sounds strange, please rephrase.

Line 175-176: Use the full names of the tree species.

Line 186-188: Use the full names of the tree species.

Line 188: Explain the term polonynas.

Line 192-193: Not possible to change now, but why was the study design not balanced?

Line 197-198: Order from plain to mountains, as you did in your description of the zones.

Line 202: Was the handheld GPS precise enough for measuring the elevation? How large is the uncertainty with this method? Please add in the manuscript.

Line 222: Why did you average across the two sampling years?

Line 228: “species richness per 100 m²“ to be consistent with the other quotations of the 100 m² plots.

Line 287-310: Could you explain a bit how the SEM disentangles direct and indirect effects?

Results:

Line 334: What does “s.e.m.” stand for? I am used to “S.E.” as abbreviation for standard error.

Discussion:

Overall: I think the red line of arguments could be a bit more consistent throughout the manuscript. For instance, I miss some discussion on the role of earthworms, which were mentioned in all parts of the manuscript except for the discussion.

Effects of grazing generally: What about the selection effect of the grazing animals? They avoid toxic plants, which you analyzed separately, but you did not include the “selective grazing” topic in the discussion.

Line 387-397: These sentences are results. Do not repeat them in the discussion. You should delete this part here and include these sentences in the results section, if they are not already in the results.

Line 430: Why “Similarly”? Isn´t the following sentence an antipode to the sentence before?

Line 454: Do you really have a larger site productivity in the higher elevated zone? If yes, could you explain this, because usually productivity decreases with elevation?

Line 458: two points at the end of the sentence

Line 458: When you use “on the other hand” you should also have “on one hand” before.

Line 469: How does this fit to Fig. 4?

Reviewer #2: This manuscript presents the results of a study across 31 grassland sites that represent an elevational and cattle density gradient in the eastern Carpathians. In particular, I appreciate that the authors present this complex issue in a way that is detailed and shows that complexity rather than oversimplifying it. There are three major take home messages: 1. Cattle grazing intensity both directly and indirectly decreases plant species richness and functional diversity and increases the proportion of the community that is considered undesirable weeds. 2. Elevation had a strong direct positive effect on plant species richness but also two negative indirect effects. 2. The effect of cattle grazing does not change with increasing elevation. In general, I find these results to be compelling and important and likely address some inconsistencies within the literature that are pointed out by the authors. However, I have two major concerns regarding the analyses presented.

1. Spatial autocorrelation – I could not find in the manuscript a discussion of how spatial autocorrelation was handled. This may have important implications for the results as the plots appear to be collocated in three different regions which are also associated with elevation and cattle density and how close they are to each other may also exert some control over species richness. It could be that I’ve just missed this – in which case I would ask that the authors highlight this a bit more prominently. In the event that I didn’t miss this – I don’t have a particular favorite way of addressing spatial autocorrelation but I’ve included a methods paper that discusses the pros and cons of some different methods (Dormann et al. Ecography). I would be happy with any of these methods as long as spatial autocorrelation is accounted for.

2. The community composition analysis using the NMDS is not well introduced in the introduction and the hypothesis is not as clear to me as the hypothesis for the SEM. I would suggest either removing it or providing a specific hypothesis that is introduced in the introduction along with the predictions for the SEM.

Minor comments:

Undesirable weeds is a tricky thing to say and I would like a description of how species were identified as undesirable. I would also maybe consider different language that isn’t so values oriented, depending on how the species were identified as undesirable maybe non-forage species would be applicable?

24 – land use should be “land use change”

26 – indirect effects should be “the indirect effects”

27 – direct should be “the direct”

34 – direct should be “the direct”

44 – I would say along an elevation gradient here instead

45 – with elevation should be “with increasing elevation”, I think throughout

58 – I would appreciate a few examples of what ecosystem functions and services you are referencing

65 – land use intensity should be “the land use intensity”

67 – I would change can to may here – or could whichever you prefer (I prefer may, � )

70-116 – I really appreciate the complexity here and how thorough you are with the predictions. I would make sure however, that each of these paragraphs has a sentence at the beginning that signals the intention of the paragraph and gives an overarching topic that isn’t linked directly to the first prediction on the list. You do this quite well with the paragraph starting on line 70.

117- 127 – I would consider moving this paragraph up to before you start your prediction paragraphs as it outlines your argument about the novelty and importance of this paper.

292 – I would say “meet the assumptions of normality…” rather than meet

Discussion – I would appreciate a paragraph that goes back to the idea introduced in the introduction about the conflicting results seen in previous studies. Do these results explain some of those conflicts?

6. PLOS authors have the option to publish the peer review history of their article (what does this mean?). If published, this will include your full peer review and any attached files.

Reviewer #1: No

Reviewer #2: No

---

## [Author Response · Author response to Decision Letter 0]

22 Aug 2020

The responses are given in the cover letter but I also copy the responses in a field here as required.

We are grateful to the editor and the reviewers for the constructive comments and suggestions, which helped to improve the manuscript. We addressed all the comments by the reviewers in the revised version of the manuscript (attached) and highlighted all changes made in manuscript and supporting information in green. 

Comments and responses:

Referees' comments and responses:

Reviewer #1: General comments: This is a well-written manuscript about an interesting topic: Direct and indirect effects of land-use intensity on plant communities across an elevational gradient. The studied plant communities were semi-natural grasslands in the Eastern Carpathians in Ukraine. This region is in fact largely under-represented in the scientific literature, as the authors state in their conclusion. Further, I like the combination of vegetation data with data from other organisms, e.g. earthworms. The dataset is therefore promising, but I have some concerns about the analysis and the presentation of the data. My largest concern is about the relation between the elevation and the grazing intensity. The authors show in the results that grazing intensity is negatively related to elevation, which is quite logic. Further, they quote that grazing intensity has large direct effects on the vegetation independent from elevation. I cannot see, in the current version of the manuscript, how the applied methods can disentangle this interrelationship. 

Response: We thank the reviewer for this positive evaluation of our manuscript and for seeing the potential in our approach. In response to the reviewer comment on how the applied methods can disentangle the direct and indirect effects, we have added more explicit statement clarifying SEM methods and direct and indirect effects: (lines 147-150) “To address questions 1 and 2 we use structural equation modelling (SEM), which allows testing simultaneous influences of multiple factors from observational data in complex systems and to distinguish direct and indirect effects of these factors [49]”; and (lines 321-341) “The direct grazing effects in our study (Fig. 1, path 1) assume that grazing intensity impacts the plant variables independently of grazing-induced variations in soil biotic and abiotic properties. The indirect effects of grazing infer that grazing intensity impacts the plant variables through the changes in bare soil exposure (Fig. 1, pathway 2→3), soil chemical properties (Fig. 1, pathway 9→8 and pathway 2→10→8), or soil biocommunity variables (Fig. 1, pathway 4→5; pathway 2→6→5; pathway 4→7→8). The direct elevation effects (Fig. 1, path 11) assume that the impact of elevation on the plant variables is independent of the elevation-induced alterations of grazing intensity and soil properties. The indirect effects of elevation in our study assumes that the plant variables vary across the elevation gradient through the changes in grazing pressure (Fig. 1, pathway 12→1), soil conditions (Fig. 1, pathway 13→8; pathway 15→3) or the follow-up alterations in soil biocommunities (Fig. 1, pathway 13→14→5; pathway 15→6→5). SEM allows for the inference of such indirect effects from observational data in complex systems by analysing the covariance structure of multiple variables [49]. At first, we constructed hypothetical model (Fig. 1) to allow for direct and indirect effects of grazing intensity, elevation, soil properties, and soil biota on plant response variables (see S1 Table for ecologically meaningful relationships that we hypothesized based on the literature).”

Reviewer #1: This leads me to another concern: I am not sure if the data presentation is optimal. While some tables in the supporting information seem to be relevant for understanding the argumentation of the data analyses (e.g. table S5 and S6), I do not see the benefit of figure 4, which shows the relationship between elevation and grazing intensity. I think more emphasize has to be laid on how this overlay of the gradients (elevation, grazing intensity and soil types as shown by Fig. S1) was disentangled.

Response: We thank the reviewer for expressing this concern on the data presentation. In the previous version of the manuscript it was not explicitly addressed in the ‘Introduction’ that we also test the plant community composition. We believe that this possibly led to the confusion regarding Fig. 4, which in fact shows plant community composition and the effects of elevation and grazing intensity on plant community composition. Therefore, we have explained this separately in the ‘Introduction’ (lines 151-152) “To test question 3 we use nonmetric multidimensional scaling (NDMS)”. Furthermore, we have added the clarifications to the caption of the Fig 4 (lines 711-712) and to ‘Discussion’ (lines 457, 502-504, 517, 548-550). We hope the reviewer finds these revisions satisfactory. 

We also agree that the effects of soil type on plant variables may be important. However, we did not explicitly measure soil types in our study. The map of the soil types (as shown by S1 Fig) is rather used to support some of our hypotheses (lines 113), methods (lines 172-174, 187-188, 200-201), and discussion (line 545).

Reviewer #1: Another question I have is concerned with the introduction: For me it is quite unusual to cite tables of the supporting information and to include a figure in the introduction. Does this fit to the journal guideline of PlosOne? Additionally, which methods were used to come up with figure 1, is it already a product of this study? Then I would handle it as such.

Overall, I recommend major revisions of this article.

Response: Thank you for raising this concern. We agree that citing figure and table in the introduction can be unusual. However we did this because Fig. 1 and S1 Table explain and detail the main hypotheses about direct and indirect effects that we test. Interpretations of SEM results must be supported not only by the data, but also based on known mechanisms. That is why we summarized a priori information (from the literature) in S1 Table and related it directly to our hypothesis (i.e., Fig. 1). Therefore, we believe that Fig. 1 and S1 Table are useful for understanding the framework and analytical approach of our study. As suggested by the editor, we refer to this issue more explicitly in the ‘Introduction’ (lines 150-151) “Fig. 1 shows the conceptual graph underlying our study and, thus, the SEM model”. In response to the reviewer comment on which methods were used to come up with figure 1, we have added the following statement (lines 73-74) “..the hypothesized mechanisms derived from literature are summarized in S1 Table and Fig. 1)”. We also clarified this in the caption of fig 1 (line 685). Furthermore, we have added the clarifications in the ‘Material and Methods’, as follows: (lines 334-337) “At first, we constructed a hypothetical model (Fig. 1) to allow for direct and indirect effects of grazing intensity, elevation, soil properties, and soil biota on plant response variables (see S1 Table for ecologically meaningful relationships that we hypothesized based on the literature)” and (lines 340-341) “Further, we tested the hypothetical model (Fig. 1) with our sampled data (Fig. 3, S6 Table)”.

We hope the reviewer finds these clarifications reasonable. 

We thank the reviewer for seeing the potential in our article.

Reviewer #1 (Minor comments, Abstract):

Line 24: most vulnerable compared to what?

Response: We have removed “most” (line 24). 

Reviewer #1: Line 26: “may depend on environmental conditions and indirect effects are rarely considered” is very vague. Could you specify what you mean?

Response: We have clarified this in lines 26-27. 

Line 29-30: The cattle grazing took place for two years? I guess you mean your sampling, you have to change the structure of the sentence. 

Response: We have revised the sentence as suggested (line 30).

Reviewer #1 (Minor comments, Introduction):

Fig. 1: Why has the elevation no influence on the soil community? Could the soil community not influence the bare soil? When was this figure produced, was it for this paper? Maybe it is already a result of the study?

Response: We did not consider the direct effects of elevation on earthworms and soil microorganisms because elevation affects the soil community mostly via changes in soil properties. Therefore, we tested the indirect effects of elevation on soil community through changes in soil properties along elevation gradient (as captured by path 13 and path 14 in Fig. 1). In response to the reviewer’s question “Could the soil community not influence the bare soil?” we have added the following statement to the ‘Material and Methods’ (lines 337-340) “Although soil biota may influence bare soil exposure, here we assumed prevalence of grazing-induced bare soil impacts on the soil community (Fig. 1, path 6) because we expected the direct effect of cattle trampling to be the most severe on bare soil exposure (Fig. 1, path 2) [25]”. 

Fig. 1 was produced specifically for the current study based on the previous literature on the topic and serves as a conceptual graph underlying our study in general and the structure of our SEM model in particular. To clarify this, we have added the statements in lines 73-74, 150-151, 334-337, 340-341, 685.

Reviewer #1: Line 133: I recommend to cite your map (figure 2) in the section about the study area.

Response: We have added the citation of the map (Fig. 2) as recommended (line 162). Fig. 2 is also cited in the lines 158, 165, 168, 180, 225.

Reviewer #1: Line 133-137: I would recommend formulating more specific research questions or hypotheses.

Response: We have formulated more specific research questions as suggested (lines 143-147)

Reviewer #1 (Methods): Line 142: “for decades” to the end of the sentence.

Response: We have moved “for decades” to the end of the sentence” (line 156).

Reviewer #1: Line 146: “significant amount of precipitation” sounds strange, please rephrase.

Response: We have replaced the phrase “significant amount of precipitation” with “high precipitation amounts” (line 160).

Reviewer #1: Line 175-176: Use the full names of the tree species.

 Line 186-188: Use the full names of the tree species.

Response: We have added the full names of the tree species throughout the manuscript as suggested (lines 175-178, 189-191, 201-204).

Reviewer #1: Line 188: Explain the term polonynas.

Response: We have explained the term polonynas as following (lines 204-207): “polonynas that are subalpine grassland landscapes in the Ukrainian part of the Eastern Carpathian Mountains, which developed above the upper forest limit within the elevation range of 1200 and 2000 m a.s.l. [51]”.

Reviewer #1: Line 192-193: Not possible to change now, but why was the study design not balanced?

Response: We agree that it would be more logical having 12 grassland sites (instead of 7) in Mountains similarly to that in the plains and foothills. We were limited in management support and facilities for field work in mountain zone. However, we believe that this did not affect our results because we tested the effects along the elevation gradient but not of the physical-geographical zone. We have added the following statement to clarify this (lines 210-212): “The unbalanced study design was not an issue in this study because we did not consider the effects of physical-geographical zones while instead we tested the effects of elevation as a continuous variable”

Reviewer #1: Line 197-198: Order from plain to mountains, as you did in your description of the zones.

Response: We have revised the sentence as suggested (lines 216-218).

Reviewer #1: Line 202: Was the handheld GPS precise enough for measuring the elevation? How large is the uncertainty with this method? Please add in the manuscript.

Response: We have added in the manuscript the accuracy of the GPS measurements (line 223) as suggested. We believe that the uncertainty in our elevation measurements (due to an error of the GPS estimated averaged positions) unlikely impacted the results of our current study because we used a single-level data set averaged for the entire grassland. The distance between the two closest study sites is ~2.18 km (measured as straight line distance between the two points on the map). 

Reviewer #1: Line 222: Why did you average across the two sampling years?

Response: In the current study we seek to understand the elevational variation rather than temporal or within-site variations. Therefore, summarizing the data at the site level (i.e. mean for the entire grassland) and further across the two sampling years was a way to create a single-level data set for each study site. To address this we have added the text to the lines 315-320.

Reviewer #1: Line 228: “species richness per 100 m²“ to be consistent with the other quotations of the 100 m² plots.

Response: We have revised the units as suggested (line 250).

Reviewer #1: Line 287-310: Could you explain a bit how the SEM disentangles direct and indirect effects?

Response: We have added the explanation on what we mean by direct and indirect effects and on how the SEM disentangles direct and indirect effects (lines 321-334): “The direct grazing effects in our study (Fig. 1, path 1) assume that grazing intensity impacts the plant variables independently of grazing-induced variations in soil biotic and abiotic properties. The indirect effects of grazing infer that grazing intensity impacts the plant variables through the changes in bare soil exposure (Fig. 1, pathway 2→3), soil chemical properties (Fig. 1, pathway 9→8 and pathway 2→10→8), or soil biocommunity variables (Fig. 1, pathway 4→5; pathway 2→6→5; pathway 4→7→8). The direct elevation effects (Fig. 1, path 11) assume that the impact of elevation on the plant variables is independent of the elevation-induced alterations of grazing intensity and soil properties. The indirect effects of elevation in our study assumes that the plant variables vary across the elevation gradient through the changes in grazing pressure (Fig. 1, pathway 12→1), soil conditions (Fig. 1, pathway 13→8; pathway 15→3) or the follow-up alterations in soil biocommunities (Fig. 1, pathway 13→14→5; pathway 15→6→5). SEM allows for the inference of such indirect effects from observational data in complex systems by analysing the covariance structure of multiple variables [49]”.

Reviewer #1 (Results): Line 334: What does “s.e.m.” stand for? I am used to “S.E.” as abbreviation for standard error.

Response: We have replaced “s.e.m.” with “S.E.” throughout the manuscript (lines 395 and 401) and Supporting Information (lines 40 and 47, Tables S3 and S5).

Reviewer #1 (Discussion): Overall: I think the red line of arguments could be a bit more consistent throughout the manuscript. For instance, I miss some discussion on the role of earthworms, which were mentioned in all parts of the manuscript except for the discussion.

Response: We agree that the discussion should be consistent with the rest of the MS. Therefore, we have added some discussion points on plant community composition (lines 502-504, 524, 532-534, 548-550), effects of soil pH (lines 537-541), impact of elevation on grazing effects (lines 559-566) and on potential explanations of inconsistencies in previous research (lines 583-609). As suggested, we have also added material on our results for earthworms and microorganisms (line 505-514) to the discussion. Additional discussions on these results are at lines 536, 546-548, 644-646. 

Reviewer #1: Effects of grazing generally: What about the selection effect of the grazing animals? They avoid toxic plants, which you analyzed separately, but you did not include the “selective grazing” topic in the discussion.

Response: We have addressed the “selection effect” in the discussion as suggested (lines 467-470).

Reviewer #1: Line 387-397: These sentences are results. Do not repeat them in the discussion. You should delete this part here and include these sentences in the results section, if they are not already in the results.

Response: We have removed this paragraph as suggested.

Reviewer #1: Line 430: Why“Similarly”? Isn´t the following sentence an antipode to the sentence before?

Response: We have added the explanations to the sentence (lines 489-490) thus connecting it to the following sentence starting with “Similarly”.

Reviewer #1: Line 454: Do you really have a larger site productivity in the higher elevated zone? If yes, could you explain this, because usually productivity decreases with elevation?

Response: In the ecological literature precipitation is frequently referred to as a proxy for site productivity (e.g., Koerner et al.). Based on that in the previous version of the MS we assumed that we might have larger site productivity in higher elevation because of higher annual precipitation (lines 453-455 in the previous version). However, in the current version of the MS we have removed this sentence from the discussion (line 526) because it would conflict with our results of lower soil organic matter with elevation, and therefore possibly lower site productivity. We thank the reviewer for bringing this up.

Koerner SE et al. Change in dominance determines herbivore effects on plant biodiversity. Nature Ecology and Evolution. 2018;2: 1925–1932.

Reviewer #1: Line 458: two points at the end of the sentence

Response: We have removed the point.

Reviewer #1: Line 458: When you use “on the other hand” you should also have “on one hand” before.

Response: We have removed “on the other hand” (line 534).

Reviewer #1: Line 469: How does this fit to Fig. 4? 

Response: We have connected the discussion to our results on plant community composition (lines 457, 502-504, 517, 548-550).

 

Reviewer #2: This manuscript presents the results of a study across 31 grassland sites that represent an elevational and cattle density gradient in the eastern Carpathians. In particular, I appreciate that the authors present this complex issue in a way that is detailed and shows that complexity rather than oversimplifying it. There are three major take home messages: 1. Cattle grazing intensity both directly and indirectly decreases plant species richness and functional diversity and increases the proportion of the community that is considered undesirable weeds. 2. Elevation had a strong direct positive effect on plant species richness but also two negative indirect effects. 2. The effect of cattle grazing does not change with increasing elevation. In general, I find these results to be compelling and important and likely address some inconsistencies within the literature that are pointed out by the authors. 

Response: We thank the reviewer for summarizing the main findings so aptly and recognizing the relevance of these results.

Reviewer #2: However, I have two major concerns regarding the analyses presented.

1. Spatial autocorrelation – I could not find in the manuscript a discussion of how spatial autocorrelation was handled. This may have important implications for the results as the plots appear to be collocated in three different regions which are also associated with elevation and cattle density and how close they are to each other may also exert some control over species richness. It could be that I’ve just missed this – in which case I would ask that the authors highlight this a bit more prominently. In the event that I didn’t miss this – I don’t have a particular favorite way of addressing spatial autocorrelation but I’ve included a methods paper that discusses the pros and cons of some different methods (Dormann et al. Ecography). I would be happy with any of these methods as long as spatial autocorrelation is accounted for. 

Response: We fully agree that the spatial autocorrelation in some variables, e.g. in elevation and in the variables correlated with elevation is to be expected because the study sites are located in three different physical-geographical zones linked to elevation. To test if further spatial autocorrelation among the study sites affected our results, we performed additional quantitative analyses (i.e., Moran’s I statistic of residuals from models for each study variable) added to the revised manuscript (lines 371-385, and S11 Table in the “Supporting Information”). These analyses showed no signiﬁcant autocorrelation among residuals for all variables suggesting that spatial autocorrelation among study sites did not affect our results. Notably, we are thankful for this important comment by the reviewer, which led us to perform these additional analyses that support our results.

Reviewer #2: 

2. The community composition analysis using the NMDS is not well introduced in the introduction and the hypothesis is not as clear to me as the hypothesis for the SEM. I would suggest either removing it or providing a specific hypothesis that is introduced in the introduction along with the predictions for the SEM. 

Response: Thank you for emphasizing this. We have provided in the “Introduction” a specific research question for the community composition analysis using the NMDS: “(3) Is plant community composition affected by elevation, cattle density, soil properties, and soil biota?” (lines 146-147) and “To test question 3 we use nonmetric multidimensional scaling (NDMS)”(lines 151-152).

Reviewer #2: Minor comments: 

Undesirable weeds is a tricky thing to say and I would like a description of how species were identified as undesirable. I would also maybe consider different language that isn’t so values oriented, depending on how the species were identified as undesirable maybe non-forage species would be applicable? 

Response: We appreciate the suggestion of the reviewer about the terminology. We would like to clarify that in our study we use the term “undesirable” to value the plant species that potentially reduce grazing efficiency and livestock production based on the following two criteria: (1) plants unpalatable to livestock including noxious weeds that are harmful, injurious, or poisonous to livestock, and (2) competitive weeds that are not poisonous and somewhat palatable low-forage-value species, which can outcompete desirable forage species, thus reducing grazing efficiency of cattle by increasing search time for food. Therefore, we decided in favour of the term “undesirable” as it has a broader meaning compared to “non-forage”, and because the term “undesirable” is widely used in rangeland, pasture and grazing management in the same context as we use it in our study. However, we have replaced the term “undesirable weeds” across the manuscript with the “undesirable species” because term “weed” itself is a plant considered undesirable in a particular condition or habitat. We have clarified the explanations to the methods on the denotation of the “undesirable species” as used in our study (lines 256-266): “Further, species were classified as undesirable for grazing (S4 Table) [52–54] if they were known to reduce grazing efficiency, forage yield, palatability and quality, therefore contributing to lower forage and animal production of grassland ecosystems [55]. The group of undesirable species includes both unpalatable species as well as competitive weeds. Unpalatable plants are those containing toxic compounds poisonous to cattle (e.g., Equisetum arvense, Ranunculus acris, Saponaria officinalis, Euphorbia sp.), or, when eaten, may cause mechanical injuries because of a spiny covering or fine hairs (e.g., Carduus crispus) [53]. Competitive weeds (e.g., some coarse tall grasses and forbs) are not toxic to cattle and somewhat palatable (e.g., Plantago sp.), but can increase in density over time and outcompete desirable forage species. In a pasture, they reduce grazing efficiency of cattle by increasing search time for high-quality food”.

Reviewer #2: 24 – land use should be “land use change” 

Response: We have added “change” to “land use” here as suggested.

Reviewer #2: 26 – indirect effects should be “the indirect effects” 

Response: We have made the corrections as suggested.

Reviewer #2: 27 – direct should be “the direct” 

Response: We have made the corrections as suggested.

Reviewer #2: 34 – direct should be “the direct” 

Response: We have made the corrections as suggested.

Reviewer #2: 44 – I would say along an elevation gradient here instead 

Response: We have changed the text here to “along an elevation gradient” as suggested.

Reviewer #2: 45 – with elevation should be “with increasing elevation”, I think throughout 

Response: We have added “increasing” as suggested throughout the text.

Reviewer #2: 58 – I would appreciate a few examples of what ecosystem functions and services you are referencing 

Response: We have added the examples of ecosystem services as suggested (lines 61-63): “..for example pollination [7], carbon storage [8–10], wildlife habitat provisioning [11–18], soil erosion control, and water ﬂow regulation [19,20]”

Reviewer #2: 65 – land use intensity should be “the land use intensity” 

Response: We have made the corrections as suggested.

Reviewer #2: 67 – I would change can to may here – or could whichever you prefer (I prefer may, � ) 

Response: We have changed “can” to “may” here as suggested.

Reviewer #2: 70-116 – I really appreciate the complexity here and how thorough you are with the predictions. I would make sure however, that each of these paragraphs has a sentence at the beginning that signals the intention of the paragraph and gives an overarching topic that isn’t linked directly to the first prediction on the list. You do this quite well with the paragraph starting on line 70. 

Response: Thank you for this positive evaluation on how we approach the description of the predictors. As suggested, we have restructured the first sentence for each paragraph (lines 86 and 103-104), thus making it clear from the beginning what the paragraph is about. 

Reviewer #2: 117- 127 – I would consider moving this paragraph up to before you start your prediction paragraphs as it outlines your argument about the novelty and importance of this paper. 

Response: We thank the reviewer for this suggestion. The mentioned paragraph (lines 126-136 in the current version of the MS) explains how to handle multiple mechanisms and the resultant indirect effects and states the gaps in a previous research. With this paragraph we make a logical connection between the previous paragraphs (on the different possible mechanisms and indirect effects, lines 75-125) to the last paragraph on what we do in our study (lines 137-152). Therefore, we decide against moving the mentioned paragraph.

Reviewer #2: 292 – I would say “meet the assumptions of normality…” rather than meet 

Response: We have made the corrections as suggested.

Reviewer #2: Discussion – I would appreciate a paragraph that goes back to the idea introduced in the introduction about the conflicting results seen in previous studies. Do these results explain some of those conflicts? 

Response: We have added a paragraph to ‘Discussion’ (lines 583-609), which discusses how our study addresses the inconsistencies in previous evidence on plant diversity responses to grazing.

---

## [Decision Letter · Decision Letter 1]

16 Oct 2020

PONE-D-20-07402R1

Direct and indirect effects of land-use intensity on plant communities across elevation in semi-natural grasslands

PLOS ONE

Dear Dr. Buzhdygan,

Thank you for submitting your manuscript to PLOS ONE. After careful consideration, we feel that it has merit but does not fully meet PLOS ONE’s publication criteria as it currently stands. Therefore, we invite you to submit a revised version of the manuscript that addresses the points raised during the review process.

The reviewers and I appreciate the careful revisions made, and there are just two, really minor points (see also reviewer #1) that should be addressed before I can make a final decision:

Lines 520 - 521: Please rephrase the sentence, e.g. "The positive effect of elevation on species number of rushes and sedges as observed in our study (S8 Table) might be attributed ..."

Line 596: Please use plural: "... along elevation gradients".

We look forward to receiving your revised manuscript.

Kind regards,

Harald Auge

Academic Editor

PLOS ONE

Reviewers' comments:

Reviewer's Responses to Questions

**Comments to the Author**

1. If the authors have adequately addressed your comments raised in a previous round of review and you feel that this manuscript is now acceptable for publication, you may indicate that here to bypass the “Comments to the Author” section, enter your conflict of interest statement in the “Confidential to Editor” section, and submit your "Accept" recommendation.

Reviewer #1: (No Response)

Reviewer #2: All comments have been addressed

2. Is the manuscript technically sound, and do the data support the conclusions?

Reviewer #1: Yes

Reviewer #2: Yes

3. Has the statistical analysis been performed appropriately and rigorously? 

Reviewer #1: Yes

Reviewer #2: Yes

4. Have the authors made all data underlying the findings in their manuscript fully available?

Reviewer #1: Yes

Reviewer #2: (No Response)

5. Is the manuscript presented in an intelligible fashion and written in standard English?

Reviewer #1: Yes

Reviewer #2: (No Response)

6. Review Comments to the Author

Reviewer #1: General comments:

I enjoyed reading this revision about the direct and indirect effects of land-use intensity on plant communities across an elevational gradient. The manuscript improved much from the first to the revised version, especially the introduction is now clearer and the discussion is more consistent with the red line of the manuscript. The authors considered all comments on an earlier version of the manuscript or responded to my suggestions. Therefore, I have no more concerns.

Two minor comments:

Line 520: The sentence should start with “in our study”.

Line 596: along elevation gradients (plural).

Reviewer #2: Many thanks to the authors for carefully addressing my comments especially as pertains to the spatial autocorrelation analysis! I'm happy to see that these results did not alter the main take home messages of this paper. I believe that this paper will make an excellent contribution to the literature and am happy with the revisions as they currently stand.

7. PLOS authors have the option to publish the peer review history of their article (what does this mean?). If published, this will include your full peer review and any attached files.

Reviewer #1: No

Reviewer #2: No

---

## [Author Response · Author response to Decision Letter 1]

19 Oct 2020

Referees' comments and responses:

Reviewer #1: General comments:

I enjoyed reading this revision about the direct and indirect effects of land-use intensity on plant communities across an elevational gradient. The manuscript improved much from the first to the revised version, especially the introduction is now clearer and the discussion is more consistent with the red line of the manuscript. The authors considered all comments on an earlier version of the manuscript or responded to my suggestions. Therefore, I have no more concerns.

Response: We thank the reviewer for this positive evaluation of our work on the revision and of our study in overall. The comments and suggestions of the both reviewers have highly improved this work.

Reviewer #1: Two minor comments:

Line 520: The sentence should start with “in our study”.

Response: We have rephrased this sentence "The positive effect of elevation on species number of rushes and sedges as observed in our study (S8 Table) might be attributed ..." (lines 520 – 521);

Reviewer #1: Line 596: along elevation gradients (plural).

Response: As suggested, in line 596 we have used plural: "... along elevation gradients".

Reviewer #2: Many thanks to the authors for carefully addressing my comments especially as pertains to the spatial autocorrelation analysis! I'm happy to see that these results did not alter the main take home messages of this paper. I believe that this paper will make an excellent contribution to the literature and am happy with the revisions as they currently stand.

Response: Thank you for this positive evaluation of our work on the revision and for seeing the potential of our paper. The comments and suggestions of the both reviewers have greatly improved the paper.

---

## [Editor Report · Decision Letter 2]

20 Oct 2020

Direct and indirect effects of land-use intensity on plant communities across elevation in semi-natural grasslands

PONE-D-20-07402R2

Dear Dr. Buzhdygan,

We’re pleased to inform you that your manuscript has been judged scientifically suitable for publication and will be formally accepted for publication once it meets all outstanding technical requirements.

Kind regards,

Harald Auge

Academic Editor

PLOS ONE
---

## [Editor Report · Acceptance letter]

12 Nov 2020

PONE-D-20-07402R2 

Direct and indirect effects of land-use intensity on plant communitiesacross elevation in semi-natural grasslands 

Dear Dr. Buzhdygan:

I'm pleased to inform you that your manuscript has been deemed suitable for publication in PLOS ONE. Congratulations! Your manuscript is now with our production department. 

Kind regards, 

on behalf of

Dr. Harald Auge 

Academic Editor

PLOS ONE